# Detecting Pulmonary Oxygen Toxicity Using eNose Technology and Associations between Electronic Nose and Gas Chromatography–Mass Spectrometry Data

**DOI:** 10.3390/metabo9120286

**Published:** 2019-11-22

**Authors:** Thijs T. Wingelaar, Paul Brinkman, Rianne de Vries, Pieter-Jan A.M. van Ooij, Rigo Hoencamp, Anke-Hilse Maitland-van der Zee, Markus W. Hollmann, Rob A. van Hulst

**Affiliations:** 1Diving and Submarine Medical Center, Royal Netherlands Navy, Rijkszee en Marinehaven, 1780 CA Den Helder, The Netherlands; 2Department of Anesthesiology, Amsterdam University Medical Center, location AMC, Meibergdreef 9, 1105 AZ Amsterdam, The Netherlands; 3Department of Pulmonology, Amsterdam University Medical Center, location AMC, Meibergdreef 9, 1105 AZ Amsterdam, The Netherlands; 4Breathomix, Pascalstraat 13H, 2811 EL Reeuwijk, the Netherlands; 5Department of Surgery, Alrijne Hospital, Simon Smitweg 1, 2353 GA Leiderdorp, The Netherlands; 6Defense Healthcare Organisation, Ministry of Defence, Herculeslaan 1, 3584 AB Utrecht, The Netherlands; 7Leiden University Medical Center, Albinusdreef 2, 2333 ZA Leiden, The Netherlands

**Keywords:** oxygen diving, two-way orthogonal partial least square regression, O2PLS, mixomics

## Abstract

Exposure to oxygen under increased atmospheric pressures can induce pulmonary oxygen toxicity (POT). Exhaled breath analysis using gas chromatography–mass spectrometry (GC–MS) has revealed that volatile organic compounds (VOCs) are associated with inflammation and lipoperoxidation after hyperbaric–hyperoxic exposure. Electronic nose (eNose) technology would be more suited for the detection of POT, since it is less time and resource consuming. However, it is unknown whether eNose technology can detect POT and whether eNose sensor data can be associated with VOCs of interest. In this randomized cross-over trial, the exhaled breath from divers who had made two dives of 1 h to 192.5 kPa (a depth of 9 m) with either 100% oxygen or compressed air was analyzed, at several time points, using GC–MS and eNose. We used a partial least square discriminant analysis, eNose discriminated oxygen and air dives at 30 min post dive with an area under the receiver operating characteristics curve of 79.9% (95%CI: 61.1–98.6; *p* = 0.003). A two-way orthogonal partial least square regression (O2PLS) model analysis revealed an R² of 0.50 between targeted VOCs obtained by GC–MS and eNose sensor data. The contribution of each sensor to the detection of targeted VOCs was also assessed using O2PLS. When all GC–MS fragments were included in the O2PLS model, this resulted in an R² of 0.08. Thus, eNose could detect POT 30 min post dive, and the correlation between targeted VOCs and eNose data could be assessed using O2PLS.

## 1. Introduction

Prolonged exposition to hyperoxia can induce pulmonary oxygen toxicity (POT) [1,2]. Aside from its importance in anaesthesia and intensive care medicine, POT is rarely a concern in general medicine. However, in diving and hyperbaric medicine, where oxygen is breathed under increased atmospheric pressures, the risk of developing POT is severe [3]. To prevent POT from occurring, oxygen exposure under increased atmospheric pressure is maintained within strict limits [4,5]. Although these limits are based on dated research and are inaccurate, no valid alternative to prevent POT has been developed [3,6,7]. Recent studies, using exhaled breath analysis by gas chromatography-mass spectrometry (GC–MS), have revealed markers of POT-associated inflammation (such as cyclohexane) and lipoperoxidation of alveolar membranes (several methyl alkanes) as compounds of interest [8,9,10].

While these volatile organic compounds (VOCs) are promising specific biomarkers for POT, GC–MS is a time and resource consuming method and therefore is not suited for point-of-care applications in diving and hyperbaric medicine [11,12,13]. In contrast, cross-reactive and gas-sensor driven electronic nose (eNose) technology fits the requirements for point-of-care applications in exhaled breath analysis [14]. However, the POT-specific molecules of interest identified by GC–MS are not represented by eNose breath profiles, which are based on the overall mixture of VOCs in exhaled breath [15,16].

Conversely, eNose technology can overcome some limitations of GC–MS, such as failure to detect unstable or reactive VOCs and molecules smaller than the lower detection threshold of GC–MS, and can recognize patterns in large clusters of data that would be regarded as background noise in GC–MS [17]. Despite these advantages, eNose sensor data can be difficult to interpret because it cannot readily be associated with any particular biological substrate or pathophysiological process [15].

Whether eNose can be employed to detect POT after hyperbaric–hyperoxic exposure is currently unknown. As eNose sensors are cross-reactive and not molecule specific, it remains to be determined whether eNose data mirror reported GC–MS findings [10]. Tailoring eNose breathprints using GC–MS profiles might be an effective way to optimize eNose accuracy for POT detection. For instance, GC–MS measurements could provide details on the biophysiological substrate of eNose signals [18].

This study aimed to determine whether eNose exhaled breath profiles, obtained before and after hyperbaric–hyperoxic exposure, are suitable biomarkers for POT and correlate to GC–MS. We hypothesized that eNose would detect POT and that eNose exhaled breath patterns could be tailored toward a model for the detection of POT-associated inflammation and lipoperoxidation by aligning eNose and GC–MS data.

## 2. Results

Twelve Navy divers (aged 36.1 ± 10.3 years; height 183.0 ± 6.2 cm; weight 90.5 ± 7.3 kg) were included in this study. All subjects completed both dives and there were no protocol violations.

### 2.1. eNose

One hundred and forty-four samples were collected and analyzed according to the protocol. The eNose optimally differentiated oxygen and air dives at 30 min post dive with an area under curve (AUC) of 79.9% (95% CI: 61.1–98.6) and a *p*-value of 0.003 (Figure 1b). At other time points, test characteristics were poor and showed no statistical significance (Figure 1).

### 2.2. Associating eNose and GC–MS Data

All eNose data (at sensor level) and the targeted VOCs were included in the two-way orthogonal partial least square regression (O2PLS) model, which resulted in a model with a R^2^ of 0.50 (Figure 2). A visual representation of the model is displayed in Figure 2, and (standardized) vectors of the regression coefficients are shown in Table 1. This model could moderately associate the VOCs with the sensor data, with an R-squared of 0.50.

An inverse of the above analysis was also conducted; addressing whether sensor data were associated with GC–MS data. This O2PLS regression model included all sensor data and all 3796 ion fragments from the GC–MS dataset [10]. This resulted in a correlation (R^2^) of 0.08. The model is displayed in Figure 3.

## 3. Discussion

This study shows that POT can be detected 30 min post dive using eNose, while at other time points post dive detection is poor and not statistically significant. A modest correlation (R^2^ = 0.50) between previously published GC–MS data and the present study was found using O2PLS regression. This means that POT-associated VOCs, generally methyl alkanes and cyclohexane, identified 3 h post dive using GC–MS, cannot be accurately detected using the sensor array of the eNose. Conversely, distinguishing data picked up by the eNose sensor arrays 30 min after oxygen diving did not corelate with the GC–MS profiles. Therefore, we could not determine which biological substrates caused these distinctive eNose patterns.

The difference in the eNose patterns between 30 min post oxygen and 30 min post air dive was statistically significant, with a *p*-value of 0.003 and an AUC of 79.9% (61.1–98.6), implying that this instrument performs moderately in distinguishing between oxygen and air dives. [19]

There are many interesting eNose studies which describe patterns in exhaled breath data. However, often the question remains which pathophysiological substrate, and consequently which VOCs, are responsible for the sensor data. Using O2PLS regression, we were able to associate GC–MS to eNose sensor data. The association between the VOCs of interest and the sensor data (R^2^ of 0.50) can be considered moderate to good [20]. The VOCs of interest seem to correlate only partially with the eNose data (Figure 2 and Table 1). This is expected, as the SpiroNose was not specifically designed to detect POT. Sensor cross-reactivity is demonstrated in Figure 3 because sensors 4, 5, and 6 showed considerable overlap. Interestingly, an association between the VOCs of interest to several sensors was found, while these sensors are responsive to different groups of molecules according to the manufacturer’s information. This further underlines cross-reactivity, and O2PLS regression allows in vivo evaluation of each sensors’ contribution to the detection of the VOCs of interest.

Furthermore, many GC–MS fragments poorly overlapped with the sensor data, which is reflected by the low R^2^ of 0.08 and the difference in the directions of the vectors plotted in Figure 3. This is surprising, as eNose sensor data entails all of the VOCs that pass through. Thus, it would make sense that the all ion fragments found by GC–MS would produce similar results. Apparently, some VOCs that generate sensor data are not picked up by GC–MS and vice versa. Additionally, in our previous study, GC–MS optimally discriminated between air and oxygen dives after 3 h post dive, while in the present study, eNose optimally discriminated between air and oxygen dives 30 min post dive [10]. This might be because eNose can detect unstable or reactive VOCs, or molecules smaller that the lower detection threshold of GC–MS. Alternatively, eNose may recognize patterns in large clusters of data that would be regarded as background noise in GC–MS analysis [17].

Although a previous study reported the detection of differences in exhaled breath composition, using eNose and GC–MS, the present study is the first to attempt associating GC–MS (ion) fragments with eNose sensor data [18]. Most studies with eNose technology are aimed at diagnosing disease or differentiating between diagnoses in a clinical environment, while GC–MS is often utilized to gain deeper insights into pathophysiological processes [11,12]. As both methods are based on different analytic principles, direct comparison of the results of these two methods can be difficult and requires complex statistical analyses [21,22].

The main strength of this study is that the eNose and GC–MS data of the exhaled breathes of divers after oxygen and air dives could be correlated using O2PLS regression. However, the correlation was moderate, although samples were collected after exposure to the same hyperbaric–hyperoxic exposure. An additional strength is the eNose sensor’s sensitivity to each VOC of interest (Table 1).

There are several limitations to this study. Firstly, as this study was conducted in a single centre, the validity of the results will have to be assessed externally. However, samples were collected according to standardized procedures by trained personnel and data were handled separately by different researchers (eNose: RdV, GC–MS: PB and TW), who were unaware of each other’s findings in order to minimize observer bias. Secondly, the experimental conditions (breathing a partial pressure of oxygen of 192.5 kPa for 1 h) were chosen to reflect those of clinical or operational applications in diving and hyperbaric medicine. Exposure to higher partial pressures of oxygen might increase the statistical correlation between eNose and GC–MS data for the detection of POT. However, it could also increase the risk of cerebral oxygen toxicity, which would be ethically unacceptable [7]. Thirdly, as our GC–MS scan range was limited from 37 to 300 Da, VOCs with a molecular mass less than 37 Da could not be identified. These small VOCs could be responsible for the difference in sensor data post dive in this study. If these small compounds could be included in the O2PLS regression, the R² was likely to increase. Lastly, with respect to exhaled breath markers, neither reference nor baseline values for these markers currently exist for either GC–MS or eNose. This is a prerequisite for any clinical or operational application of these two methods for the detection of POT. Whether a pre dive measurement remains necessary to detect POT post dive, or whether only a post dive measurement suffices remains to be elucidated.

### Direction of Future Research

This study assessed the association between the responses of individual sensors in an eNose to POT-associated VOCs detected by GC–MS. For practical applications of eNose in this field, increases in sensitivity or specificity will be required to make valid diagnoses. This could be achieved by modifying eNose to equip it with more accurate sensors for the detection of POT-associated VOCs. After which, similar tests could be conducted to evaluate the accuracy of future iterations of eNose and this cycle could be repeated until the instrument meets the accuracy demanded. Eventually, this could remove the barrier to replacing resource consuming GC–MS with low-cost eNose for exhaled breath analysis.

## 4. Materials and Methods

A double-blinded randomized cross-over trial was conducted at the Royal Netherlands Navy Diving Medical Center, Den Helder, the Netherlands, and was carried out in accordance with the recommendations of the Ethics Committee of the University of Amsterdam and the principles of the Declaration of Helsinki. Written consent was obtained from all subjects. The protocol was approved by the Medical Ethical Committee of the University of Amsterdam (Reference: 2017.183) and the Surgeon General of the Ministry of Defense. The study was registered at the Dutch Trial Register (ID: NTR6547).

### 4.1. Study Design

Healthy non-smoking male Navy divers made two dives, separated by a week, of sixty minutes to 192.5 kPa, equivalent to a depth of nine meters of sea water [10]. One dive was made with pure oxygen, while the other was made with compressed air (20.7% oxygen). The subjects and researchers were blinded to the gas they breathed.

Participants were not exposed to hyperbaric or hyperoxic conditions for at least 72 h prior to the start of the study. During the study and the day before hyperbaric exposure, no strenuous physical exercise (including sports) was performed. To avoid affecting exhaled breath profiles, divers had to fast for 1 h before the first measurement and were only allowed to drink water. Between the third and fourth measurement, food (bread and jelly) was provided and divers were encouraged to eat to prevent changes in metabolism caused by fasting [23].

Pre dive measurements were performed 30 min before hyperbaric exposure. Post dive, the measurements were performed at 30 min and then each full hour thereafter until the 4 h post dive. eNose measurements were performed before GC–MS sample collection. See Figure 4 for reference.

### 4.2. eNose

This study utilized the SpiroNose which enables real time analysis of exhaled breath profiles [24]. A detailed description of the procedures and sensor stability verifications has been published previously [24,25]. This eNose comprised seven different cross-reactive metal-oxide semiconductor sensor arrays (nodes), with four sensors per array (Figaro Engineering Inc, Arlington Heights, IL, USA) and is capable of sampling both exhaled breath and surrounding air. For every breath profile, sensor data were normalized to that of the most stable sensor array (sensor 2).

Subjects were asked to rinse their mouth thoroughly with water before measurements. Subsequently, exhaled breath analysis was performed twice with a 2 min interval between each analysis. All participants were instructed to perform five tidal breaths followed by a single inspiratory capacity manoeuvre up to total lung capacity, a 5-s breath-hold, and a slow (<0.4 L/s) maximal expiration to reach residual volume. Exhaled breath was measured in real time by the eNose sensors. Obtained data were sent directly to and stored on an online server. Raw sensor data were downloaded from the server and prepared for analysis using Matlab (MathWorks, Natick, MA, USA). Further details on the data handling are published elsewhere [25].

### 4.3. GC–MS Dataset

The GC–MS dataset used for comparison and alignment with eNose data was published previously [10]. Seven VOCs of interest were reconstructed from this dataset: cyclohexane, 2,4-dimethylhexane, 2-methylnonane, 3-[(1,1-dimethylethoxy)methyl]heptane, nonanal, decane and decanal. The largest difference in the intensity of VOCs of interest after dives with oxygen and compressed air was 35%, which occurred 3 h post dive. Additionally, all ion fragments from this dataset (*n* = 3796) were also used to associate eNose and GC–MS data.

### 4.4. Statistical Analysis

All statistical analyses were performed using the R software package (version 3.6.1, R Foundation for Statistical Computing, Austria), including surrogate variable analysis (SVA version 3.32.1), Methods for the Behavioral, Educational, and Social Sciences (MBESS version 4.6.0) and the MixOmics package (MixOmics version 6.8.0).

Partial least squares discriminant analysis (PLS-DA) at each timepoint (*t* = 0, *t* = ½, *t* = 1, *t* = 2, *t* = 3, *t* = 4) was utilized to evaluate whether the eNose could distinguish between air and oxygen dives. To evaluate the discriminative power of the obtained model, the model-specific derived PLS-DA components 1 and 2 were used to compute receiver operating characteristics (ROC) and area under curve (AUC), including 95% confidence intervals and *p*-values. A *p*-value of <0.05 was considered statistically significant.

### 4.5. Associating eNose and GC–MS Data

Although both eNose and GC–MS can be utilized to examine exhaled metabolites, the profiles do not necessarily show a 100% overlap or perfect correlation [12]. A two-way orthogonal partial least square regression (O2PLS) is considered valid to compare two large-scale (‘omics’) datasets, while compensating for the risk of systemic variation [21].

Associating eNose and GC–MS data, for both the reconstructed VOCS as well as all ion fragments, was done in accordance with previous published O2PLS-based methods [21,22]. The explained variance (R^2^) was used to assess the validity of the model. The vector of regression components, i.e., the contribution of each sensor to the detection of a VOC, was used to determine the contribution of each sensor to the detection of a specific VOC. Inverse modelling, to determine whether eNose sensor signals could be explained by GC–MS data, was also conducted using O2PLS.

## 5. Conclusions

The present study assessed the ability of eNose to detect POT and examined whether eNose data correlate with GC–MS data using two-way orthogonal partial least square regression (O2PLS). While the ‘off the shelf’ eNose only provided moderate to poor correlation with POT-associated VOCs, the results of this study should enable researchers to tailor eNose so that it achieves the sensitivity and specificity of GC–MS. Future studies will be aimed at developing a high sensitivity and low cross-reactivity sensor array for the detection of POT.

## Figures and Tables

**Figure 1 metabolites-09-00286-f001:**
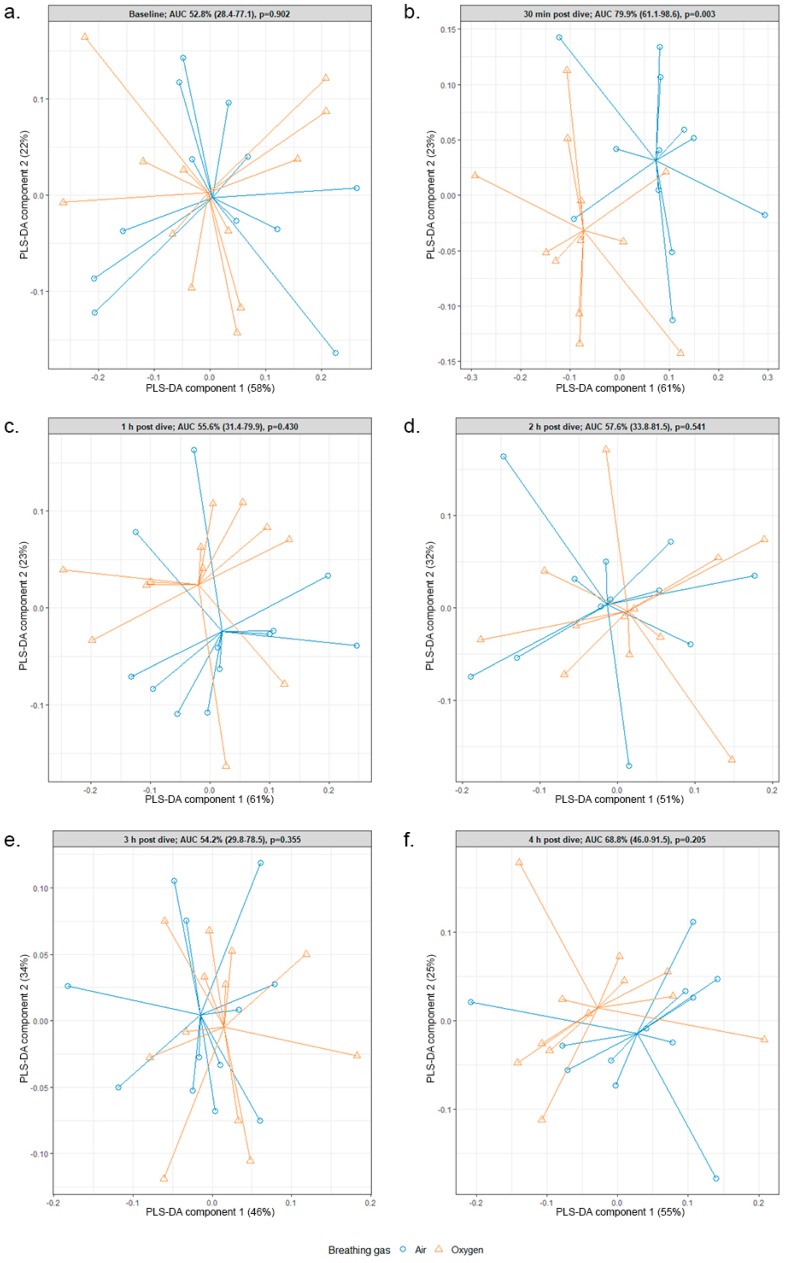
Partial least squares discriminant analysis of the electric nose (eNose) data. Oxygen dives are displayed in orange (triangles) and air dives in blue (circles). (**a**) 30 min pre dive; (**b**) 30 min post dive; (**c**) 1 h post dive; (**d**) 2 h post dive; (**e**) 3 h post dive; (**f**) 4 h post dive. The area under curve (AUC), including 95% confidence interval, as well as *p*-value are displayed in the title of the plot window. Note that as a result of the Mix0mics package and individual correction, the spider plots are mirrored from each other.

**Figure 2 metabolites-09-00286-f002:**
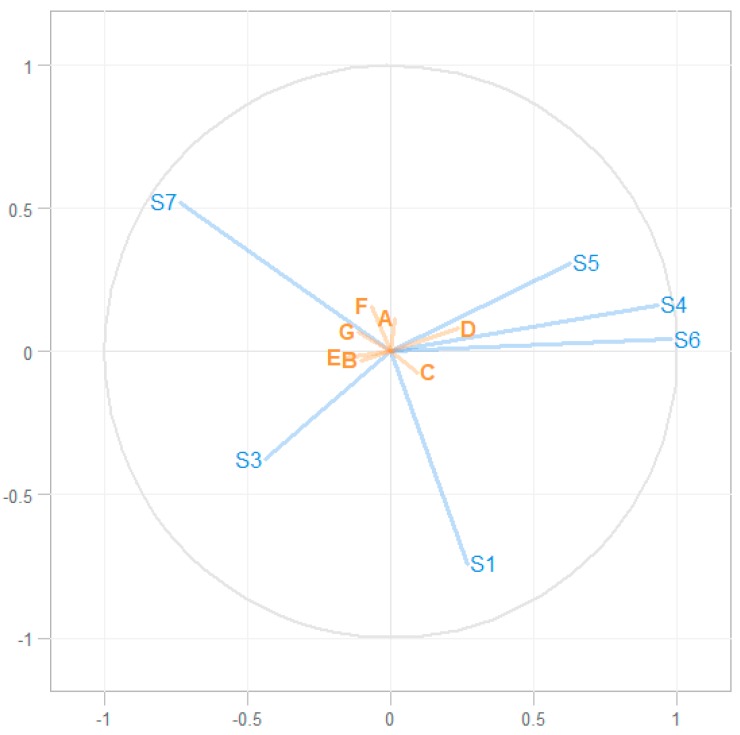
Association between the eNose sensor data and target VOCs. In blue: the six eNose sensors included in the two-way orthogonal partial least square regression (O2PLS) modelling (numbered). In orange: the seven pulmonary oxygen toxicity (POT)-related exhaled volatile organic compounds (VOCs); A: Cyclohexane, B: 2,4-Dimethylhexane, C: 3-Methylnonane, D: 3-[(1,1-dimethylethoxy)methyl]heptane, E: Nonanal, F: Decane, G: Decanal. This O2PLS regression model has an R^2^ of 0.50.

**Figure 3 metabolites-09-00286-f003:**
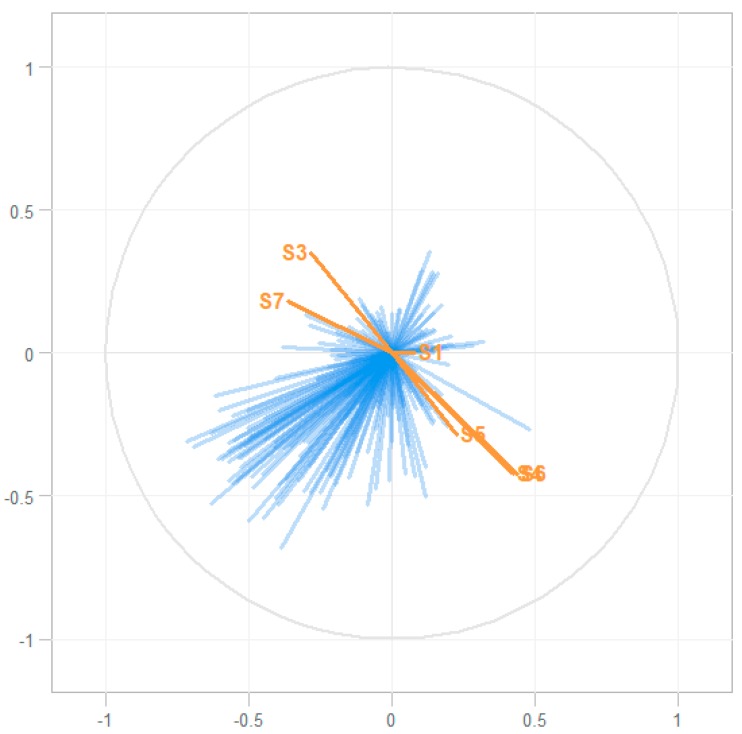
Association between all ion fragments and eNose sensor data. Sensors are displayed in orange. Again, sensor 2 was used for calibration and is not displayed. Ion fragments (*n* = 3796) detected with GC–MS are displayed in blue. The R^2^ of this O2PLS model is 0.08.

**Figure 4 metabolites-09-00286-f004:**
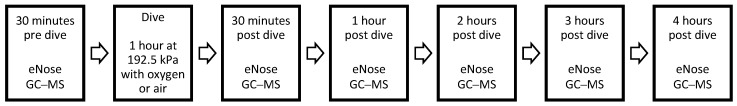
Overview of the study design and data collection. Both study days were identical, the only difference being the type of exposure during the dive (either 100% oxygen or compressed air).

**Table 1 metabolites-09-00286-t001:** Vectors of standardized regression coefficients. Rows: the six eNose sensors included in the O2PLS modelling. Note that sensor array 2 was used to calibrate the signals and therefore not included in the O2PLS. Columns: the seven POT-related exhaled VOCs; A: Cyclohexane, B: 2,4-Dimethylhexane, C: 3-Methylnonane, D: 3-[(1,1-dimethylethoxy)methyl]heptane, E: Nonanal, F: Decane, G: Decanal.

	A	B	C	D	E	F	G
S1	−0.065	0.010	0.048	−0.029	0.001	−0.092	−0.046
S3	−0.056	0.031	0.018	−0.076	0.037	−0.060	−0.012
S4	0.013	−0.035	0.024	0.080	−0.053	−0.011	−0.031
S5	0.050	−0.032	−0.012	0.077	−0.039	0.051	0.006
S6	0.007	−0.031	0.026	0.072	−0.049	−0.016	−0.032
S7	0.057	0.011	−0.063	−0.020	0.032	0.099	0.066

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
