# Peer review of "Detecting Pulmonary Oxygen Toxicity Using eNose Technology and Associations between Electronic Nose and Gas Chromatography–Mass Spectrometry Data"

_metabolites, 2019, doi:10.3390/metabo9120286_

Round 1

Reviewer 1 Report

This paper, describes first, a discriminant analysis based on Partial least square to evaluate whether an Enose could distinguish between air and oxygen dives. Then,  whether the E-nose data are correlated to the GC-MS data is examined by using Two-way Orthogonal Partial least square regression.

The first part of the “Results section” is quite understandable as the projections of the 24 samples on the two component 1 and 2 calculated by PLS regression show a clearly separation of air or oxygen at 30 min post dives. However, the signification of indicators like the Area under Curve (AUC) or the P-Value, which are used to conclude to the E-nose ability, must be described in the paper.

The second part of this section contains many errors making the comprehension very difficult. It is necessary to rewrite this part.

Here are some comments:

What is the interest of studying the correlation between gas sensors and VCO compounds? It is well known that a metal oxide gas sensor is not specific and that it responds to many VOCs gases Figure 3 is identical to figure 2. There is no any of the 3796 ion fragments in figure 3. Is it make sense to associate 6 sensors data in one hand and 3796 Ion fragments data in the other hand?

My decision is major revision especially on the results and discussion.

Author Response

Thank you for reviewing the manuscript. We would like to address your comments point-by-point. For clarity we have also included your comments (in italic text).

This paper, describes first, a discriminant analysis based on Partial least square to evaluate whether an Enose could distinguish between air and oxygen dives. Then, whether the E-nose data are correlated to the GC-MS data is examined by using Two-way Orthogonal Partial least square regression.

The first part of the “Results section” is quite understandable as the projections of the 24 samples on the two component 1 and 2 calculated by PLS regression show a clearly separation of air or oxygen at 30 min post dives. However, the signification of indicators like the Area under Curve (AUC) or the P-Value, which are used to conclude to the E-nose ability, must be described in the paper.

Our initial paper did not describe the input for calculating AUC and a description of its usage. We have rewritten the second paragraph in the statistical analysis section of the methods section to address this. It now reads (page 8, lines 234-239):

“Partial least squares discriminant analysis (PLS-DA) at each timepoint (t=0, t=½, t=1, t=2, t=3, t=4) was utilized to evaluate whether the eNose could distinguish between air and oxygen dives. To evaluate the discriminative power of the obtained model, the model specific derived PLS-DA components 1 and 2 were used to compute receiver operating characteristics (ROC) and area under curve (AUC), including 95% confidence intervals and p-values. A p-value of < 0.05 was considered statistically significant.”

The second part of this section contains many errors making the comprehension very difficult. It is necessary to rewrite this part.

This was probably caused by discrepancies between the methods and results section in the initial manuscript. In the revised paper we have made several additions to the methods section (page 8, sections 4.3 to 4.5), as well as some alterations to the discussion as outlined below. We feel this enhanced comprehension of the presented data.

What is the interest of studying the correlation between gas sensors and VCO compounds?

There are many very interesting eNose studies which describe patterns in exhaled breath data. However, often the question remains which pathophysiological substrate, and consequently which VOCs, are responsible for the sensor data. 

In this study we aimed to correlate VOCs to sensor data through O2PLS regression. This could help answering the question stated above. As figure 2 and table 1 shows, each sensors contribution in the detection of a single VOC of interest can determined though O2PLS regression. To our knowledge, this has not been published before.

This combination of exhaled breath modalities from eNose and GC-MS could answer both questions.  We have further emphasized this in the manuscript on page 6 (line 123-126): “There are many interesting eNose studies which describe patterns in exhaled breath data. However, often the question remains which pathophysiological substrate, and consequently which VOCs, are responsible for the sensor data. Using O2PLS regression we were able to associate GC-MS to eNose sensor data.”

It is well known that a metal oxide gas sensor is not specific and that it responds to many VOCs gases

That is correct, and you bring up a very important point. Sensor manufacturers often state to which types of compounds their sensors are responsive to, but it is very difficult for researchers to test this in vivo. In our experience from previous studies, we could not always explain eNose sensor data by ‘backtracking’ which types of compounds could have triggered responses based on manufacturers information. As stated above; O2PLS regression enables researchers to correlate VOCs to sensor data.

To expand on this topic, we have added the following lines to the discussion (page 6, lines 130-133):

“Interestingly, an association of VOCs of interest to several sensors was found while these sensors are responsive to different groups of molecules according to the manufacturer’s information. This further underlines cross-reactivity, and O2PLS regression allows in vivo evaluation of each sensors’ contribution to the detection of VOCs of interest.”

Figure 3 is identical to figure 2. There is no any of the 3796 ion fragments in figure 3.

You are correct. It seems figure 3 was incorrectly displayed in the manuscript. This has been corrected in the revised document on page 5. Our sincerest apologies for this deficiency.

Is it make sense to associate 6 sensors data in one hand and 3796 Ion fragments data in the other hand?

As the correlation between VOCs of interest and sensor data was modest (R² 0.50), we concluded that either our GC-MS analysis was incomplete or that eNose sensors could pick up exhaled breath data that GC-MS does not (such as very small or very reactive compounds). In the first case, associating all ion fragments to the sensor data could tell if specific groups or compounds were missed, for instance by regarding VOCs as background noise due to a relatively low signal to noise ratio.

In the second scenario, we would have found results as presented in this manuscript. Figure 3, which is now correctly added in the manuscript, shows sensors data which can poorly be explained by the ion fragments (for instance 3 and 7). Additionally, cross-reactivity is shown as many ion fragments are picked up by sensor 5 and 7, but as you correctly mentioned earlier, this is a widely known fact (but now presented visually).

To answer the question; we feel is makes sense to associated all ion fragments to sensor data, as eNose sensors are exposed to the full spectrum of exhaled breath, and not just the VOCs of interest. To further stress this, we have added the following lines to the discussion (page 6, lines 135-138):

 “This is surprising, as eNose sensor data entails all VOCs that pass through, it would make sense that the all ion fragments found by GC-MS would produce similar results. Apparently, some VOCs that generate sensor data are not picked up by GC-MS and vice versa. “

Reviewer 2 Report

The paper describes the use of eNose for detecting changes in VOC compositions related to exposure to oxygen under increased atmospheric pressure, which can induce pulmonary oxygen toxicity. Correlation between eNose data and GC-MSD data is also addresses. While the subject is of great interest, the results suffer on two aspects. The eNose sensors were not specifically designed for the compounds of interest in this study and the MSD scanning range in GC-MSD analysis was not properly choose, limiting the ability of technique to completely identify the VOCs. Despite this, the paper is of high scientific quality.

Particular comments:

- define ROC in abstract

- lines 117-119: one orange point (0.1/0.024) is in the middle of the blue ones (and, consequently, one blue point (-0.1/-0.024) is in the middle of the orange ones); is there any explanation for this?

- line 129: As mentioned in reference 10, the scan range in GC-MSD analysis was 37-300 Da. This range could be shifted to lower values, since volatile compounds do not reach 300 Da and lighter compounds with molecular mass below 37 (e.g. CO, ammonia, formaldehyde, methane, ethane, etc.) are not detected by MS. These latter compounds could be instead detected by eNose.

Author Response

We would like to address your comments point-by-point. For clarity we have also included your comments (in italic text).

The paper describes the use of eNose for detecting changes in VOC compositions related to exposure to oxygen under increased atmospheric pressure, which can induce pulmonary oxygen toxicity. Correlation between eNose data and GC-MSD data is also addresses. While the subject is of great interest, the results suffer on two aspects. The eNose sensors were not specifically designed for the compounds of interest in this study and the MSD scanning range in GC-MSD analysis was not properly choose, limiting the ability of technique to completely identify the VOCs. Despite this, the paper is of high scientific quality.

Thank you for your kind words.

The GC-MS scanning range is an important limitation of our study which has now been added to the manuscript (see also below). As molecule specific sensors are currently lacking, and possibly never see the light of day using metal oxide gas sensor technology currently used in many eNose platforms, we feel this paper supports the ongoing discussion how to tailor eNose technology for specific purposes.

Though O2PLS regression we were able to analyse whether the eNose was able to pick up our VOCs of interest in vivo, with background noise, instead of testing sensors in a laboratory setup, and which sensors are contributing the most. In future iterations of our studies, we plan to replace or add sensors to the eNose platform and repeat these experiments, and evaluate the discriminative power of the customized eNose.

- define ROC in abstract

This has been added to the abstract (page 1, line 30).

- lines 117-119: one orange point (0.1/0.024) is in the middle of the blue ones (and, consequently, one blue point (-0.1/-0.024) is in the middle of the orange ones); is there any explanation for this?

The orange and blue dots/points in the spider plots of the PLS-DA are mirrored from one another. This is the result of the visual representation of the PLS-DA (the plotIndiv function from the Mix0mics package), with individual correction.

This has been added to the caption of figure 1. “Note that as a result of the Mix0mics package and individual correction, the spider plots are mirrored from each other.”

- line 129: As mentioned in reference 10, the scan range in GC-MSD analysis was 37-300 Da. This range could be shifted to lower values, since volatile compounds do not reach 300 Da and lighter compounds with molecular mass below 37 (e.g. CO, ammonia, formaldehyde, methane, ethane, etc.) are not detected by MS. These latter compounds could be instead detected by eNose.

This is an important limitation to our study, and we have added it to the limitations as listed in the discussion (page 6, line 164-168): “Thirdly, as our GC-MS scan range was limited from 37 to 300 Da, VOCs with a molecular mass less than 37 Da could not be identified. These small VOCs could be responsible for the difference in sensor data post dive in this study. If these small compounds could be included in the O2PLS regression, the R² is likely to increase.”

Additionally, we have corrected two typographical errors (line 57: than instead of that; line 248: superscripted the square) and corrected the syntax of two references (the brackets of ref 10 at line 140 and of ref 12 at line 243).

Round 2

Reviewer 1 Report

Dear  Authors,

the manuscript is revised according to the recommendations. It is now suitable for publication. 

Best regards